# Maturity-Onset Diabetes of the Young (MODY) in Portugal: Novel *GCK*, *HNFA1* and *HNFA4* Mutations

**DOI:** 10.3390/jcm9010288

**Published:** 2020-01-20

**Authors:** Maria I. Alvelos, Catarina I. Gonçalves, Eduarda Coutinho, Joana T. Almeida, Margarida Bastos, Maria L. Sampaio, Miguel Melo, Sofia Martins, Isabel Dinis, Alice Mirante, Leonor Gomes, Joana Saraiva, Bernardo D. Pereira, Susana Gama-de-Sousa, Carolina Moreno, Daniela Guelho, Diana Martins, Carla Baptista, Luísa Barros, Mara Ventura, Maria M. Gomes, Manuel C. Lemos

**Affiliations:** 1CICS-UBI, Health Sciences Research Centre, University of Beira Interior, 6200-506 Covilhã, Portugal; 2C4-UBI, Cloud Computing Competence Centre, University of Beira Interior, 6200-501 Covilhã, Portugal; 3Serviço de Endocrinologia, Diabetes e Metabolismo, Centro Hospitalar Universitário de Coimbra, 3000-075 Coimbra, Portugal; 4Unidade de Endocrinologia Pediátrica, Serviço de Pediatria Médica, Departamento de Pediatria, Centro Hospitalar Universitário de Lisboa Norte, 1649-035 Lisboa, Portugal; 5Faculdade de Medicina, Universidade de Coimbra, 3000-548 Coimbra, Portugal; 6Unidade de Endocrinologia Pediátrica, Serviço de Pediatria, Hospital de Braga, 4710-243 Braga, Portugal; 7Unidade de Endocrinologia Pediátrica, Serviço de Pediatria Ambulatória, Hospital Pediátrico de Coimbra, Centro Hospitalar e Universitário de Coimbra, 3000-602 Coimbra, Portugal; 8Serviço de Endocrinologia e Diabetes, Hospital Garcia de Orta, 2805-267 Almada, Portugal; 9Consulta de Pediatria/Patologia endócrina, Centro Hospitalar do Médio Ave, Unidade de V. N. Famalicão, 4761-917 Vila Nova de Famalicão, Portugal; 10School of Medicine, University of Minho, 4710-057 Braga, Portugal

**Keywords:** diabetes, MODY, genetics, mutation

## Abstract

Maturity-onset diabetes of the young (MODY) is a frequently misdiagnosed type of diabetes, which is characterized by early onset, autosomal dominant inheritance, and absence of insulin dependence. The most frequent subtypes are due to mutations of the *GCK* (MODY 2), *HNF1A* (MODY 3), and *HNF4A* (MODY 1) genes. We undertook the first multicenter genetic study of MODY in the Portuguese population. The *GCK*, *HNF1A,* and *HNF4A* genes were sequenced in 46 unrelated patients that had at least two of the three classical clinical criteria for MODY (age at diagnosis, family history, and clinical presentation). The functional consequences of the mutations were predicted by bioinformatics analysis. Mutations were identified in 23 (50%) families. Twelve families had mutations in the *GCK* gene, eight in the *HNF1A* gene, and three in the *HNF4A* gene. These included seven novel mutations (*GCK* c.494T>C, *GCK* c.563C>G, *HNF1A* c.1623G>A, *HNF1A* c.1729C>G, *HNF4A* c.68delG, *HNF4A* c.422G>C, *HNF4A* c.602A>C). Mutation-positive patients were younger at the time of diagnosis when compared to mutation-negative patients (14.3 vs. 23.0 years, *p* = 0.011). This study further expands the spectrum of known mutations associated with MODY, and may contribute to a better understanding of this type of diabetes and a more personalized clinical management of affected individuals.

## 1. Introduction

Maturity-onset diabetes of the young (MODY) is a monogenic form of diabetes characterized by early onset (usually under the age of 25 years), autosomal dominant inheritance and absence of insulin dependence during a variable period of time [1].

MODY is estimated to account for about 1–2% of all cases of diabetes [2]. However, the high costs of genetic testing and the frequent overlap of clinical features between MODY and the more frequent types of diabetes (i.e., types 1 and 2) have resulted in a significant underdiagnosis of MODY [3]. A correct molecular diagnosis is an important aspect in patient management because it can help select the most appropriate treatment, provide a prognosis for the course of the disease, alert to the existence of associated malformations, and allow for genetic counselling [4].

There are several subtypes of MODY caused by mutations in at least 14 known genes (*HNF4A*, *GCK*, *HNF1A*, *PDX1*, *HNF1B*, *NEUROD1*, *KLF11*, *CEL*, *PAX4*, *INS*, *BLK*, *ABCC8*, *KCNJ11*, *APPL1*), which have in common a primary defect in insulin secretion associated with pancreatic beta cell dysfunction [5]. However, heterozygous mutations in the *GCK* (glucokinase) (MODY 2), *HNF1A* (hepatocyte nuclear factor 1 alpha) (MODY 3), and *HNF4A* (hepatocyte nuclear factor 4 alpha) (MODY 1) genes are the most frequent, and together they explain over 95% of the known genetic causes of MODY [6].

The relative frequencies of MODY subtypes show variations according to the countries where the studies took place. For example, MODY 3 is the most common subtype in the United Kingdom, The Netherlands, Denmark, and Norway, but MODY 2 is the most common in Germany, Austria, Poland, the Czech Republic, Italy, Greece, and Spain [6]. These differences may be explained by the use of different selection criteria for patients for genetic testing [6].

The aim of this study was to identify the underlying genetic mutations in a group of Portuguese patients with clinically suspected MODY.

## 2. Experimental Section

### 2.1. Subjects

Forty-six families with suspected MODY cases were referred by physicians based at six pediatric and adult diabetes clinics serving the northern, central, and southern regions of mainland Portugal, from 2012 to 2019. Patients were referred for genetic testing due to clinical suspicion of MODY based on at least two of the following three criteria: (a) diagnosis of diabetes under the age of 25 years in at least one family member; (b) autosomal dominant inheritance pattern through at least three generations, or the existence of at least two first-degree relatives with diabetes; (c) ability to control diabetes without insulin treatment for at least two years, or significant levels of serum C-peptide, or absence of pancreatic autoantibodies. Whenever possible, other affected and non-affected family members were studied. The control population consisted of 500 healthy volunteers recruited among blood donors. All patients and controls were Caucasian Portuguese. Written informed consent was obtained from all subjects and the study was approved by the local research ethics committee (Faculty of Health Sciences, University of Beira Interior, Ref: CE-FCS-2012-010).

### 2.2. Genetic Studies

Genomic deoxyribonucleic acid (DNA) was extracted from peripheral blood leucocytes [7] and used with custom-designed primers for polymerase chain reaction (PCR) amplification of the coding regions and exon–intron boundaries of the *GCK*, *HNF1A,* and *HNF4A* genes. Bidirectional sequencing of the PCR products was performed using a CEQ DTCS sequencing kit (Beckman Coulter, Fullerton, CA, USA) and an automated capillary DNA sequencer (GenomeLab TM GeXP, Genetic Analysis System; Beckman Coulter). Genomic sequence variants identified in patients were searched for in the Genome Aggregation Database (gnomAD) [8] in order to assess their frequency in the general population. Novel missense variants were screened for in a panel of 200 healthy Portuguese volunteers (400 alleles) using sequence-specific restriction enzymes or allele-specific PCR, to exclude the possibility that these were common population-specific polymorphisms. For one variant found in a healthy control, the screening was extended to 500 controls (1000 alleles). Variants that were found to be present at an allele frequency higher than 0.1% in either gnomAD or in the Portuguese control population were excluded from further analysis. Computational functional prediction analysis was performed to evaluate the impact of the sequence variants on protein function, using SIFT [9], PolyPhen-2 [10], Mutation Taster [11], and Human Splicing Finder [12] programs. Sequence variants were analyzed by VarSome [13] and classified according to the American College of Medical Genetics and Genomics (ACMG) guidelines [14]. Mutation nomenclature was based on the cDNA reference sequences for the *GCK* (NM_000162.5), *HNF1A* (NM_000545.5), and *HNF4A* (NM_175914.3) genes.

### 2.3. Statistical Analysis

The clinical characteristics of patients with and without identified mutations (i.e., mutation-positive and mutation-negative patients) were compared. Mean age at time of diagnosis, body mass index (BMI), glycated hemoglobin (A1c), and serum C-peptide were compared by two-tailed Student’s t-tests. The proportions of patients in each group with positive family history (defined as autosomal dominant inheritance pattern through at least three generations, or the existence of at least two first-degree relatives with diabetes), typical clinical presentation and course (defined as ability to control diabetes without insulin treatment for at least two years, or significant levels of serum C-peptide, or absence of pancreatic autoantibodies), and presence of all three classical MODY clinical criteria, were compared by a two-tailed Fisher’s exact test. Statistical significance was set at *p* < 0.05.

## 3. Results

Rare heterozygous sequence variants were identified in 23 (50%) of 46 families (Figure 1a–d). Twelve occurred in the *GCK* gene, eight in the *HNF1A* gene, and three in the *HNF4A* gene (Figure 2). Three variants were found to be recurrent, thus the total number of unique variants was 20. These consisted of 16 missense, two nonsense, one frameshift, and one synonymous variant.

Thirteen variants have already been reported in MODY patients, namely *GCK* c.130G>A [15], *GCK* c.386G>A [16], *GCK* c.544G>A [17], *GCK* c.556C>T [18], *GCK* c.698G>A [19], *GCK* c.757G>C [20], *GCK* c.1099G>A [21], *GCK* c.1268T>A [22], *HNF1A* c.425C>T [23], *HNF1A* c.475C>T [24], *HNF1A* c.511C>T [23], *HNF1A* c.521C>T [25], and *HNF1A* c.607C>A [26].

Seven variants (*GCK* c.494T>C, *GCK* c.563C>G, *HNF1A* c.1623G>A, *HNF1A* c.1729C>G, *HNF4A* c.68delG, *HNF4A* c.422G>C, *HNF4A* c.602A>C) have not previously been reported in patients with MODY and were found to be absent or very rare (≤0.0001) in the gnomAD population database [8] (Table 1). These variants were not found in healthy Portuguese controls, except for *HNF1A* c.1729C>G, which was identified in 1/1000 alleles. These novel variants were shown to cosegregate with diabetes in additional family members, except for *HNF4A* c.68delG, because DNA from relatives was not available for testing. All variants were predicted to be deleterious by at least one of four bioinformatics programs (SIFT [9], PolyPhen-2 [10], Mutation Taster [11], or Human Splicing Finder [12]) (Table 1). The synonymous variant in *HNF1A* (c.1623G>A, p.Gln541Gln) was predicted to affect RNA splicing, by the Mutation Taster [11] and Human Splicing Finder [12] bioinformatics programs.

According to strict ACMG classification criteria [14], *HNF1A* c.521C>T, *HNF1A* c.1623G>A, and *HNF1A* c.1729C>G were classified as variants of uncertain significance and all others were classified as either pathogenic or likely pathogenic variants (Table 1).

An additional synonymous variant (*HNF4A* c.711G>A, p.Ala237Ala) was identified in one patient. Although this variant was found to be very rare in the gnomAD population database (allele frequency 3/282800 = 0.00001), it was found in several Portuguese normal controls (allele frequency 5/980 = 0.005) and was therefore considered as a population-specific polymorphism with no relation to the disorder.

The clinical characteristics of MODY patients with identified mutations are summarized in Table 2. The comparison between mutation-positive and mutation-negative patients (Table 3) showed that mutation-positive patients were significantly younger at the time of diagnosis (14.3 vs. 23.0 years, *p* = 0.011). No differences were observed regarding family history, clinical presentation, or course of the diabetes (Table 3). No differences were observed for BMI, A1c, or serum C-peptide (data not shown). The majority (*n* = 12) of mutation-positive patients presented only two of the three classical clinical criteria for MODY (Table 3).

## 4. Discussion

This study of 46 families with clinically suspected MODY revealed the presence of a genetic cause in 23 (50%) families. *GCK*, *HNF1A,* and *HNF4A* heterozygous mutations were found in 12 (26%), eight (17%), and three (7%) families, respectively.

The frequency of mutations in clinically suspected MODY cases varies across different European countries, such as Germany/Austria (97%), Spain (89%), Italy (70%), Greece (66%), Denmark (49%), the Czech Republic (48%), the Netherlands (39%), Norway (31%), the United Kingdom (27%), and Poland (7%) [6]. The differences observed between different countries are most likely explained by the different selection criteria used for genetic testing [6]. Ours is the first study in the Portuguese population and shows that the mutation frequency (50%) lies within the range for European populations [6].

The most frequent genetic defects in our population were *GCK* mutations (MODY 2), followed by *HNF1A* mutations (MODY 3). This is consistent with the distribution in other southern European populations [6]. *HNF4A* mutations (MODY 1) were rarer, as expected according to their known contribution to MODY [6]. Other MODY subtypes are even rarer and therefore were not searched for in our group of patients.

The 23 identified mutations consisted of 20 unique mutations and three recurrent mutations (i.e., occurring in more than one family). Of these 20 different mutations, 13 have already been reported in MODY [15,16,17,18,19,20,21,22,23,24,25,26]. The remaining seven mutations (*GCK* c.494T>C, *GCK* c.563C>G, *HNF1A* c.1623G>A, *HNF1A* c.1729C>G, *HNF4A* c.68delG, *HNF4A* c.422G>C, *HNF4A* c.602A>C) have not previously been reported, and are therefore novel mutations.

Bioinformatics analysis, using sequence conservation and structure-based algorithms, predicted that the identified mutations are highly likely to affect protein function. Most mutations in this study were missense mutations. These results are consistent with studies in other European populations where missense mutations were also predominant [22,26]. Missense mutations may result in alterations of secondary structures, affecting protein stability or resulting in the loss of important catalytic domains. Two nonsense and one frameshift mutations were identified—these may result in premature termination of the encoded protein or in nonsense-mediated RNA decay [27]. A synonymous mutation in *HNF1A* (c.1623G>A, p.Gln541Gln), involving the last nucleotide of exon 8, was predicted to affect RNA splicing, although no RNA was available from the patient to demonstrate this experimentally. In addition to the computational prediction results, the absence or rarity of all these mutations in large population databases [8] is highly suggestive that these are indeed pathogenic mutations associated with the disorder, rather than common benign polymorphisms.

In 23 (50%) of the families, no genetic defect was identified in the *GCK*, *HNF1A*, or *HNF4A* genes. This rate of mutation-negative cases is similar to that observed in other European populations [6] and could have several explanations. First, it may be due to the occurrence of mutations in genes that were not analyzed in the study. Although the *GCK* (MODY 2), *HNF1A* (MODY 3), and *HNF4A* (MODY 1) genes are responsible for the vast majority of genetic causes of MODY [6], there are other rarer MODY subtypes caused by mutations in genes not analyzed in this study, which could have been analyzed by next-generation sequencing methods [28]. However, these other subtypes have been shown to represent together only a small proportion (<5%) of cases or to be associated with specific extra-pancreatic features that were not present in our patients [29]. Second, mutations may rarely be located in the promoter or deep intronic regions, or result from large deletions of the genes that would not be detected by conventional sequencing [30]. Finally, mutation-negative genetic tests may be explained by the occurrence of phenocopies. This occurs when suspected MODY cases represent other types of diabetes (e.g., type 1 or 2) that can also occur in young individuals and/or coexist with other affected family members, thus mimicking the phenotype of MODY, but with no monogenic cause. As diabetes is a relatively common disease, the existence of several affected family members does not necessarily imply an inherited cause of diabetes [31]. Furthermore, the fact that mutation-negative patients in our study were diagnosed at a later age suggests that these are less likely to be associated with a genetic cause.

The inclusion criteria for patients in this study were less stringent than the classical clinical triad for MODY (i.e., diagnosis under the age of 25 years, autosomal dominant inheritance, and clinical presentation and course of the diabetes) [1], because patients do not always exhibit all the typical clinical characteristics of MODY. This occurs because family history is sometimes incomplete or unknown and vertical transmission through three generations is not always possible to confirm, patients may have de novo mutations (i.e., not present in the parents), asymptomatic hyperglycemias can remain undiagnosed beyond the age of 25 years, and the presence of pancreatic autoantibodies (which are typical for type 1 diabetes) has also been reported in MODY [32]. As our study included cases that fulfilled only two of the three above-mentioned criteria, this allowed the genetic diagnosis of 12 patients that would have otherwise remained undiagnosed, although at the expense of a higher proportion of mutation-negative cases. Several studies have proposed the selection of cases for genetic testing based on biomarkers and clinical features associated with a higher probability of having a genetic defect [33,34,35]. These prediction models show high sensitivity, although with relatively low positive predictive values that result in even higher proportions of mutation-negative cases. It is likely that the decision about which patients should undergo genetic testing, based on their prior likelihood of having a MODY mutation, will ultimately depend on the cost-effectiveness of the genetic testing [36].

In conclusion, this is the first Portuguese multicenter genetic study of MODY patients. In 46 families with clinically suspected MODY, mutations were found in 23 (50%) families, including seven *GCK*, *HNF1A*, and *HNF4A* mutations that have not previously been reported. Several mutations were identified in cases that only partially fulfilled the classical clinical criteria for MODY. The study results may contribute to a better understanding of the pathogenesis of the most common subtypes of MODY and to a more personalized approach to patients’ treatment, follow-up, and genetic counselling.

## Figures and Tables

**Figure 1 jcm-09-00288-f001:**
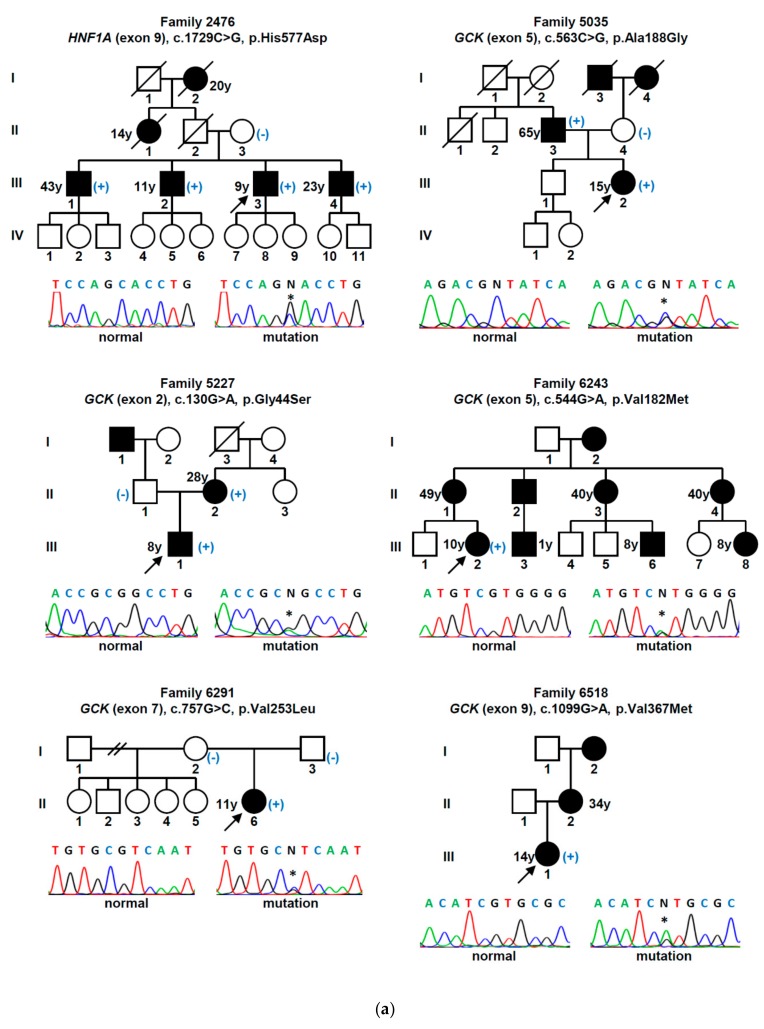
(**a**–**d**) Families with identified maturity-onset diabetes of the young (MODY) mutations. Filled symbols represent patients with diabetes, open symbols represent unaffected individuals. Squares, circles, and diamond symbols denote males, females, and unspecified, respectively. Numbers within symbols indicate additional siblings with the same phenotype. Oblique lines through symbols represent deceased individuals. Arrows indicate the index cases. The age of diagnosis of diabetes (y, years), when known, is presented. The presence (+) or absence (–) of the mutation, when known, is presented. The chromatograms of the DNA sequence for normal individuals and for patients with mutations (asterisks) are presented below each pedigree.

**Figure 2 jcm-09-00288-f002:**
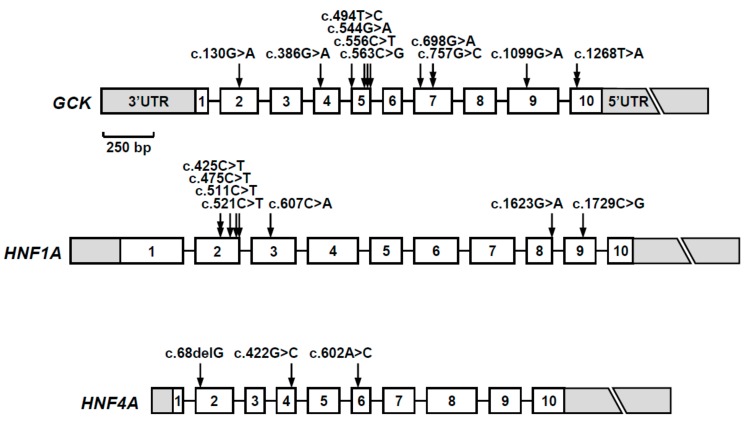
Schematic representation of the *GCK*, *HNF1A,* and *HNF4A* genes and positions of the identified mutations. Numbered boxes represent exons and lines represent introns (not to scale). UTR, untranslated regions. A 250 base-pair (bp) scale is shown.

**Table 1 jcm-09-00288-t001:** Rare sequence variants identified in MODY patients.

Gene (a)	Nucleotide Change	Amino Acid Change	Mutation Type	Population Allele Frequency (gnomAD/Portuguese Controls)	Computational Programs That Support a Pathogenic Effect (b)	Classification (ACMG Criteria) (c)	Previously Reported (Reference)
*GCK*	c.130G>A	p.Gly44Ser	missense	0	SIFT, PPh-2, MT	L. Pathogenic (PM1, PM2, PM5, PP2, PP3)	Yes [15]
	c.386G>A	p.Cys129Tyr	missense	0	SIFT, PPh-2, MT	L. Pathogenic (PM1, PM2, PP2, PP3)	Yes [16]
	c.494T>C	p.Leu165Pro	missense	0/0 (d)	SIFT, PPh-2, MT	L. Pathogenic (PM1, PM2, PP2, PP3, PP5)	No
	c.544G>A	p.Val182Met	missense	0	SIFT, PPh-2, MT	L. Pathogenic (PM1, PM2, PP2, PP3, PP5)	Yes [17]
	c.556C>T	p.Arg186*	nonsense	0	MT	Pathogenic (PVS1, PM1, PM2, PP3, PP5)	Yes [18]
	c.563C>G	p.Ala188Gly	missense	0/0 (h)	SIFT, PPh-2, MT	L. Pathogenic (PM1, PM2, PM5, PP2, PP3)	No
	c.698G>A	p.Cys233Tyr	missense	0	SIFT, PPh-2, MT	L. Pathogenic (PM1, PM2, PP2, PP3)	Yes [19]
	c.757G>C	p.Val253Leu	missense	0	SIFT, PPh-2, MT	L. Pathogenic (PM1, PM2, PM5, PM6, PP2, PP3)	Yes [20]
	c.1099G>A	p.Val367Met	missense	0	SIFT, PPh-2, MT	L. Pathogenic (PM1, PM2, PP2, PP3, PP5)	Yes [21]
	c.1268T>A	p.Phe423Tyr	missense	0	SIFT, PPh-2, MT	L. Pathogenic (PM1, PM2, PP2, PP3, PP5)	Yes [22]
*HNF1A*	c.425C>T	p.Ser142Phe	missense	0	SIFT, PPh-2, MT	L. Pathogenic (PM2, PP1, PP2, PP3, PP5)	Yes [23]
	c.475C>T	p.Arg159Trp	missense	0.0000039	SIFT, PPh-2, MT	L. Pathogenic (PM1, PM2, PM5, PP2, PP3, PP5)	Yes [24]
	c.511C>T	p.Arg171*	nonsense	0	MT	Pathogenic (PVS1, PM2, PP3, PP5)	Yes [23]
	c.521C>T	p.Ala174Val	missense	0.0001951	MT	VUS (PP2, PP3, BS2)	Yes [25]
	c.607C>A	p.Arg203Ser	missense	0	SIFT, PPh-2, MT	L. Pathogenic (PM1, PM2, PM5, PP2, PP3)	Yes [26]
	c.1623G>A	p.Gln541Gln	synonymous	0/0 (e)	MT, HSF	VUS (PM2, PP1, PP3)	No
	c.1729C>G	p.His577Asp	missense	0.0001429/0.001 (i)	SIFT, MT	VUS (PP1, PP2, PP3, BS2)	No
*HNF4A*	c.68delG	p.Gly23Alafs*81	frameshift	0	MT	Pathogenic (PVS1, PM2, PP3)	No
	c.422G>C	p.Arg141Pro	missense	0/0 (f)	SIFT, MT	L. Pathogenic (PM1, PM2, PP1, PP2, PP3)	No
	c.602A>C	p.His201Pro	missense	0/0 (g)	SIFT, PPh-2, MT	L. Pathogenic (PM1, PM2, PP2, PP3)	No

(a) Reference sequences are *GCK* (NM_000162.5), *HNF1A* (NM_000545.5), and *HNF4A* (NM_175914.3). (b) SIFT, sorting tolerant from intolerant; PPh-2, PolyPhen-2; MT, Mutation Taster; HSF, Human Splicing Finder. (c) American College of Medical Genetics and Genomics (ACMG) criteria [14] were used to classify each variant as Pathogenic, Likely (L) Pathogenic, or Variant of Uncertain Significance (VUS), based on the evidence of pathogenicity (very strong (PVS1), strong (PS1–4), moderate (PM1–6), or supporting (PP1–5)). (d–g) Determined in 200 Portuguese controls (400 alleles) using allele-specific PCR. (h) Determined in 200 Portuguese controls (400 alleles) using HpyCH4III restriction enzyme. (i) Determined in 500 Portuguese controls (1000 alleles) using allele-specific PCR.

**Table 2 jcm-09-00288-t002:** Clinical characteristics of MODY patients with identified mutations.

Patient ID	Sex/Age (yrs)	Age at Diagnosis (yrs)	Family History (a)	Presenting Signs and Symptoms	BMI (kg/m^2^)	A1c (%)	C-Peptide (ng/mL) (b)	Abs	Treatment	Complications	Last A1c (%)	Mutation
2476	M/28	9	Yes	Asymptomatic	21.8 (*)	7.2	0.10	Yes	Insulin	Retinopathy, neuropathy, nephropathy (kidney transplant)	5.2	*HNF1A* c.1729C>G (p.His577Asp)
5035	F/48	15	No	Asymptomatic	19.4 (*)	n/a	n/a	n/a	No	No	6.5	*GCK* c.563C>G (p.Ala188Gly)
5227	M/9	8	No	Asymptomatic	17.5	6.4	0.84	No	No	No	6.7	*GCK* c.130G>A (p.Gly44Ser)
6243	F/22	10	Yes	Asymptomatic	22.3 (*)	n/a	7.20	No	OHA	No	6.6	*GCK* c.544G>A (p.Val182Met)
6291	F/12	11	No	Weight loss	17.9	6.0	4.80	No	No	No	6.0	*GCK* c.757G>C (p.Val253Leu)
6518	F/15	14	Yes	Polyuria/polydipsia	25.5	6.5	4.50	No	OHA	No	6.4	*GCK* c.1099G>A (p.Val367Met)
6856	F/16	6	No	Asymptomatic	15.5	6.6	0.82	No	OHA	No	6.3	*GCK* c.494T>C (p.Leu165Pro)
6866	M/33	3	Yes	Asymptomatic	25.2 (*)	6.6	2.52	n/a	No	No	6.2	*GCK* c.1268T>A (p.Phe423Tyr)
6955	F/34	34	Yes	Asymptomatic	26.2	6.4	3.40	No	OHA	No	5.9	*HNF1A* c.521C>T (p.Ala174Val)
7004	F/32	31	Yes	Gestational diabetes	20.7 (*)	6.1	0.80	No	No	No	6.1	*GCK* c.1268T>A (p.Phe423Tyr)
7018	F/31	16	Yes	Polyuria/polydipsia	28.9	9.0	1.00	No	Insulin	Retinopathy	6.7	*HNF4A* c.422G>C (p.Arg141Pro)
7034	M/20	14	Yes	Asymptomatic	23.4 (*)	6.5	2.39	n/a	No	No	6.1	*GCK* c.386G>A (p.Cys129Tyr)
7071	F/39	21	Yes	Gestational diabetes	19.1 (*)	n/a	0.70	No	No	No	6.4	*HNF4A* c.68delG (p.Gly23Alafs *81)
7113	M/7	4	No	Asymptomatic	17.4 (*)	6.4	1.17	No	No	No	6.7	*GCK* c.757G>C (p.Val253Leu)
7396	F/13	11	Yes	Weight loss	16.2	8.7	1.80	No	Insulin	No	5.9	*HNF1A* c.425C>T (p.Ser142Phe)
7422	M/27	25	No	Asymptomatic	22.7	n/a	1.70	No	OHA + Insulin	No	5.1	*HNF1A* c.607C>A (p.Arg203Ser)
7451	F/39	18	Yes	Asymptomatic	25.8	6.4	2.30	No	OHA	No	7.3	*GCK* c.698G>A (p.Cys233Tyr)
7467	F/14	14	No	Asymptomatic	22.2	8.0	1.37	No	OHA	No	8.0	*HNF1A* c.511C>T (p.Arg171 *)
7613	M/7	6	No	Asymptomatic	15.6	6.9	0.90	No	No	No	6.7	*GCK* c.556C>T (p.Arg186 *)
7629	F/32	18	Yes	Polyuria/polydipsia	22.9 (*)	n/a	1.50	No	Insulin	No	5.6	*HNF1A* c.425C>T (p.Ser142Phe)
7646	M/35	17	Yes	Polyuria/polydipsia	19.3 (*)	n/a	0.10	No	Insulin	No	5.5	*HNF1A* c.475C>T (p.Arg159Trp)
7690	F/18	14	No	Asymptomatic	30.7	6.8	3.60	No	OHA	No	7.7	*HNF1A* c.1623G>A (p.Gln541Gln)
7797	F/12	11	Yes	Polyuria/polydipsia	24.1	12.0	2.22	No	OHA + Insulin	No	8.1	*HNF4A* c.602A>C (p.His201Pro)

(a) Autosomal dominant inheritance pattern through at least three generations, or the existence of at least two first-degree relatives with diabetes. Yrs, years. (b) Normal values > 0.8 ng/mL. M, male; F, female; BMI, body mass index (*) At last evaluation. A1c, glycated hemoglobin; Abs, positivity for at least one pancreatic autoantibody; OHA, oral hypoglycemic agents; n/a, not available.

**Table 3 jcm-09-00288-t003:** Comparison of clinical characteristics of mutation-positive and mutation-negative patients.

	Mutation-Positive (*n* = 23)	Mutation-Negative (*n* = 23)	*p*-Value (c)
Age at diagnosis (mean ± SD) (y)	14.3 ± 7.7	23.0 ± 13.2	0.011*
Family history (a) (n, %)	14 (60.9)	15 (65.2)	1.000
Clinical presentation and course (b) (n, %)	22 (95.7)	22 (95.7)	1.000
Presence of 2 inclusion criteria (n, %)	12 (52.2)	16 (69.6)	0.365
Presence of 3 inclusion criteria (n, %)	11 (47.8)	7 (30.4)	0.365

(a) Autosomal dominant inheritance pattern through at least three generations, or the existence of at least two first-degree relatives with diabetes. (b) Ability to control diabetes without insulin treatment for at least two years, or significant levels of serum C-peptide, or absence of pancreatic autoantibodies. (c) Mean age compared by two-tailed Student´s t-test (*statistically significant), all other parameters compared by two-tailed Fisher’s exact test. SD, standard deviation; y, years; n, number.

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
