# Peer review of "Maturity-Onset Diabetes of the Young (MODY) in Portugal: Novel GCK, HNFA1 and HNFA4 Mutations"

_jcm, 2020, doi:10.3390/jcm9010288_

Round 1

Reviewer 1 Report

I enjoyed reading this manuscript where the authors have investigated a cohort of MODY families in Portugal to investigate already known and identify new SNPs. 46 families were investigated and a total of 21 SNPs identified of which 13 are already known and 8 are not yet reported.

1) Line 64. Additional word 'and'

2) The authors mention the term "genetic defects" and I am not sure this is the most appropriate term. A SNP is not necessarily a defect.

3) Some of the chromatograms are not convincing, this may be due to their size and not being in colour. Consider having these in appendix in a larger form maybe. Often the baseline has risen and this is obscuring the potential heterozygous position.

4) Can a reference be added to the end of line 198?

5) Lines 234 to 236 are difficult to understand. By “absence and rarity” are the authors referring to the 13 already known SNPs or the 8 newly identifies ones, even then I’m not sure the sentence makes sense.

6) Line 248 “negative genetic tests”. What are the authors referring too? No test are negative here, the only results possible are that a SNP has been identified or not. This is not a negative test.

7) Line 251. The global prevalence of diabetes among adults over 18 years of age has risen from 4.7% in 1980 to 8.5% in 2014, I do not think it is yet a “highly prevalent disease”.

8) Line 270. This sentence I think has an error as it is not clear.

9) I would expect in a specific population, even though there are not 100's of people, to see that some SNPs come up multiple times. It seems as though each SNP was only see in individuals. I'm not therefore some of the significance of the SNPs. It may be that this work is too preliminary and requires a functional approach.

Reviewer 2 Report

In the paper 'Maturity-Onset Diabetes of the Young (MODY) in Portugal: Novel GCK, HNFA1 and HNFA4 mutations', Alvelos and colleagues provide the results of a screening for mutations in the genes GCK, HNF1A, and HNF4A. The article is generally well-written and the results are clear and appropriately contextualized to other literature. While the results are not novel per se, they do have merit as confirmation and reference for MODY-1, 2, and 3 in a population which has not been studied before. 

The major limitations with the study are the narrow selection of genes included in the analysis, the use of PCR over NGS methodology, and the lack of samples from some of the families. Furthermore, patient selection and screening in controls should be elaborated further (see below). 

Please find some specific comments below. 

Introduction: 

The introduction is well-written and provides an overview of the common MODY types for the uninitiated, but could use shortening given that much of it is spent repeating well known facts about MODY whereas the salient points regarding population/regional differences in MODY prevalence, which are the justification for the study, are swamped. 

Experimental section: 

It is unclear where the patients were selected from. The authors say multicenter, but it is unclear how many centers and how large a geographical area this represents? Were the patients from general diabetes clinics or pediatric clinics? Over how long a period were the patients collected? Were the patients referred to the authors for testing or did the authors select the patients (i.e. did the authors assess the criteria or did a referring physician)? How many patients were reviewed or referred in total to yield the 46 families? Were all patients ethnically Portugese? 

The sentence on the screening in controls is unclear to me. What does the use of 'unreported' variants refer to? Novel variants? If yes, why did the authors choose not to screen the healthy controls in the same way as cases? And what do the authors mean by 'excluded'? 

As stated above, it is a major limitation that only three genes were assessed rather than taking a more agnostic approach and using panel sequencing or WES/WGS. Beyond this, the assessment of variants is sound. 

Results: 

The authors write that the variants cosegregated with diabetes except for two, but by my count, nine families only had DNA from one individual and I would assume that makes it impossible to gage whether the variant cosegregates with disease? 

I would encourage the authors to add information on prevalence of the variants in the controls to the table also so that it is clear whether and in how many controls the variants were screened/found. 

I would also encourage the authors to double check the information on the c.711G>A variant in HNF4A. I was unable to find the variant in Gnomad. Perhaps this is due to the reference sequence? 

I find the His577Asp variant in HNF1A quite curious. It is relatively common in the European population but has not been reported in previous MODY literature according to the authors' findings. In the family in the present study, 4/4 carriers with the mutation are diabetic, but unfortunately, none of the 11 family members from generation IV have been studied. The variant has been reported in a previous population-wide screening (DOI: 10.1038/ng.2794) in a non-diabetic individual. This study quite nicely highlights that it is difficult to attribute effect sizes to MODY variants when there is inadequate data from population-based screenings and that the selection of only "extreme" phenotypes invariably leads to ascertainment bias. Just want to bring this study to the authors' attention if they are unfamiliar with it.

The authors could consider whether to include the statistical methods (table 3) in the text also. 

Discussion: 

Overall, I think the discussion does a nice job of contextualizing the findings. 

Personally, I think it is difficult to gage the likelihood of possible explanations for negative MODY findings (the authors write "Finally, a more likely explanation for the negative genetic tests is the occurrence of phenocopies.") given that many studies of MODY prevalence use quite limited methods (like this study) which make it hard to assess the other explanations, but i recognize that this is personal opinion. 

Figures: 

There seems to be an error in figure 2 where c.422G>C appears twice on the HNF4A panel, and c.711G>A not at all. 

Several of the chromatograms in figure 1 are very blurry and should be revised. 

Round 2

Reviewer 2 Report

The authors have adequately addressed the issues brought up in the peer review and I have no further comments.